# Transcriptional Landscape of *Waddlia chondrophila* Aberrant Bodies Induced by Iron Starvation

**DOI:** 10.3390/microorganisms8121848

**Published:** 2020-11-24

**Authors:** Silvia Ardissone, Aurélie Scherler, Trestan Pillonel, Virginie Martin, Carole Kebbi-Beghdadi, Gilbert Greub

**Affiliations:** Institute of Microbiology, Lausanne University Hospital and Lausanne University, Bugnon 48, 1011 Lausanne, Switzerland; Silvia.Ardissone@chuv.ch (S.A.); aurelie.scherler1@gmail.com (A.S.); Trestan.Pillonel@chuv.ch (T.P.); Virginie.Martin2@chuv.ch (V.M.); Carole.Kebbi-Beghdadi@chuv.ch (C.K.-B.)

**Keywords:** *Chlamydiales*, *Waddlia**chondrophila*, aberrant bodies, iron deprivation, persistence, transcriptomics, RNA-sequencing

## Abstract

Chronic infections caused by obligate intracellular bacteria belonging to the *Chlamydiales* order are related to the formation of persistent developmental forms called aberrant bodies (ABs), which undergo DNA replication without cell division. These enlarged bacteria develop and persist upon exposure to different stressful conditions such as β-lactam antibiotics, iron deprivation and interferon-γ. However, the mechanisms behind ABs biogenesis remain uncharted. Using an RNA-sequencing approach, we compared the transcriptional profile of ABs induced by iron starvation to untreated bacteria in the *Chlamydia*-related species *Waddlia*
*chondrophila*, a potential agent of abortion in ruminants and miscarriage in humans. Consistent with the growth arrest observed following iron depletion, our results indicate a significant reduction in the expression of genes related to energy production, carbohydrate and amino acid metabolism and cell wall/envelope biogenesis, compared to untreated, actively replicating bacteria. Conversely, three putative toxin-antitoxin modules were among the most up-regulated genes upon iron starvation, suggesting that their activation might be involved in growth arrest in adverse conditions, an uncommon feature in obligate intracellular bacteria. Our work represents the first complete transcriptomic profile of a *Chlamydia*-related species in stressful conditions and sets the grounds for further investigations on the mechanisms underlying chlamydial persistence.

## 1. Introduction

Members of the *Chlamydiae* phylum are highly diverse species of obligate intracellular Gram-negative bacteria, currently classified in more than 10 family-level lineages [1]. The well-studied *Chlamydiaceae* family includes two major human pathogens, *Chlamydia trachomatis*, one of the most prevalent sexually-transmitted bacterial pathogens [2], and *Chlamydia pneumoniae*, responsible for respiratory diseases [3]. Bacteria belonging to the other families are commonly referred to as ‘*Chlamydia*-related’ or ‘*Chlamydia*-like’ organisms [4]. First isolated from an aborted bovine foetus, *Waddlia chondrophila* belongs to the *Waddliaceae* family [5] and is a possible agent of abortion in ruminants [6,7,8]. In humans, several studies revealed a possible role of *W. chondrophila* in adverse pregnancy outcomes [9,10,11,12] and tubal infertility [13]. In addition, *W. chondrophila* might be implicated in lower respiratory tract infections such as bronchiolitis and pneumonia [14,15].

All described chlamydiae species share a unique biphasic developmental cycle alternating between two morphologically distinct forms, the infectious non-dividing elementary bodies (EBs) and the non-infectious dividing reticulate bodies (RBs) [16,17]. Following entry into a host cell through endocytosis or phagocytosis, EBs reside in a cytoplasmic vacuole, named inclusion, where they differentiate into replicative RBs. This transition involves the decondensation of the bacterial genome and the activation of a specific transcriptional program [18,19,20,21,22]. After several cycles of division through binary fission, RBs redifferentiate into EBs, which are released by host cell lysis or extrusion [23]. Upon exposure to stressful conditions, chlamydiae can develop into a third form, called aberrant bodies (ABs), a reversible stage associated to persistence. Numerous stressful conditions have been reported to induce ABs, including β-lactam antibiotics [24], phosphonic acid antibiotics [25,26], interferon-γ [27] and nutrients starvation [28]. Iron deprivation following deferoxamine or 2,2′-bipyridyl (BPDL) treatment also leads to the formation of ABs in vitro [29], a predictable observation since chlamydiae are strongly dependent on iron availability [30,31,32,33]. Interestingly, spontaneously occurring *W. chondrophila* ABs were observed in vitro in a human endometrial cell line overexpressing lactoferrin (an iron-binding glycoprotein), which likely restricts bacterial access to this important nutrient [34]. In vivo, ABs were observed by electron microscopy in the genital tract of non-treated *C. trachomatis* infected women [35]. Iron availability is a physiological parameter that fluctuates during the menstrual cycle in women, and these fluctuations might play a role in chronic infections by *Chlamydiales* [36,37].

Chlamydial iron homeostasis is complex and tightly linked to the regulation of gene expression (reviewed in [38]). Chlamydiae can acquire iron from the host by recruiting to the inclusion transferrin—containing vesicles from the slow recycling endocytic pathway [39]. It was also previously shown that *C. trachomatis* uses the ATP-binding cassette (ABC) transport system encoded by *ytgABCD* to import iron from the inclusion lumen [40,41,42,43].

While stress stimuli inducing the formation of ABs are well described, the molecular mechanisms leading to the development of enlarged chlamydiae remain poorly understood. Previous studies have investigated differences in gene expression and protein levels in ABs induced by distinct stressful conditions compared to untreated bacteria by qRT-PCR, microarray or immunoblotting, but no shared regulatory mechanisms could be identified (reviewed in [44]). This is consistent with our recent observation that different stress stimuli induce the formation of morphologically distinct ABs in *W. chondrophila* [29]. Variations in the transcriptional response of ABs induced by different stress stimuli could be due to methodological limitations, as qRT-PCR and immunoblot are low-throughput methods whereas microarrays, despite providing high-throughput screenings, show a restricted ability to quantify lowly and very highly expressed genes, and are not suitable for the analysis of highly similar sequences [45]. To overcome these limitations, RNA sequencing (RNA-seq) was recently used for transcriptome profiling of *C. trachomatis* in response to iron starvation [46].

The transcriptional response of *Chlamydia* species to different stressful conditions does not seem to be conserved, but the response to the same stress might be conserved in different chlamydial species. Only two studies investigated the transcriptional profile of EBs and RBs in *Chlamydia*-related bacteria [20,47], and to our knowledge no data are available on their response to stressful conditions. *Chlamydia*-related species possess much larger genomes and more versatile metabolic capabilities than *Chlamydia* species [48,49], which makes them particularly relevant for the investigation of the transcriptional response to adverse conditions. As the presence of ABs seems to be associated to persistent/recurrent chlamydial infections [50,51], it is crucial to understand whether the formation of ABs upon exposure to analogous stress stimuli is associated to a similar transcriptional response in distantly related chlamydial species.

In this study, we performed a comparative transcriptomic analysis of *W. chondrophila* RBs and ABs induced by the iron chelator BPDL. Our results show a significant decrease in expression of genes related to energy production, carbohydrate and amino acid metabolism and cell wall/envelope biogenesis in ABs, compared to actively replicating RBs, which is consistent with a growth arrest induced by stressful conditions. Understanding the molecular mechanisms involved in the development of persistent chlamydial forms is pivotal to fight chronic chlamydial infections. Investigating the transcriptional response to specific stress conditions in multiple *Chlamydiae* species will undoubtedly help unravel the mechanisms of persistence.

## 2. Material and Methods

### 2.1. Cell Culture and Bacterial Strains

HEp-2 cells (ATCC^®^ CCL-23^TM^) were grown in Dulbecco’s Modified Eagle Medium (DMEM, PAN-Biotech, Aidenbach, Germany) supplemented with 10% Fetal Bovine Serum (FBS, Gibco, Thermo Fisher Scientific, Waltham, MA, USA) in 75 cm^2^ flasks or in 24-well plates at 37 °C in presence of 5% CO_2_. *Acanthamoeba castellanii* (ATCC^®^ 30010^TM^) was cultured in 25 cm^2^ flasks containing 10 mL of peptone-yeast extract-glucose (PYG) medium at 25 °C. *W. chondrophila* (ATCC^®^ VR-1470^TM^) was co-cultivated with *A. castellanii* in 25 cm^2^ flasks with 10 mL of PYG broth at 32 °C. After seven days of co-culture, bacteria were collected by filtering the suspension through a 5-μM-pore filter to eliminate amoebal cysts and trophozoites. The filtrate was then diluted 10 times in PYG to infect *A. castellanii* and 100 times in DMEM to infect HEp-2 cells (MOI ~15).

### 2.2. Infection Procedure and 2,2′-bipyridyl Treatment

The day before infection, cells were seeded in 25 cm^2^ flasks (2 × 10^6^ cells per flask) or 24-well plates (2.5 × 10^5^ cells per well) containing glass coverslips, and incubated overnight at 37 °C with 5% CO_2_ until infection. Confluent cell cultures were then infected with *W. chondrophila* at a final dilution of 1:100 (MOI ~15), 1:500 or 1:1000 in DMEM. Infected HEp-2 cells were centrifuged at 1790× *g* for 10 min at room temperature and incubated for 15 min at 37 °C with 5% CO_2_. To remove non-internalized bacteria, cells were washed once with phosphate buffer saline (PBS) before adding fresh DMEM supplemented with 10% FBS.

The iron chelator 2,2′-bipyridyl was purchased from Sigma-Aldrich (Saint-Louis, MO, USA) and solubilized in 100% ethanol. For immunofluorescence experiments, BPDL was added 2, 4, 8 or 16 h post infection (hpi) at a final concentration of 50, 75 or 100 μM. Infected cells were fixed 24 hpi. For RNA extraction, infected HEp-2 cells were treated with 75 μM BPDL at 8 hpi and harvested 24 hpi (16 h treatment with BPDL). Untreated infected HEp-2 cells were harvested 24 (RBs) or 72 hpi (EBs).

### 2.3. Immunofluorescence Staining

*W. chondrophila* infected HEp-2 cells grown on glass coverslips were fixed 24 hpi with ice-cold methanol for 5 min and washed three times with PBS. Coverslips were then incubated for at least 2 h in blocking solution (PBS, 1% bovine serum albumin, 0.1% saponin, 0.04% sodium azide) at 4 °C. For immunofluorescence staining, glass coverslips were incubated with rabbit anti-*W. chondrophila* antibodies (1:5000 dilution) [52] for 2 h in a humidified chamber. Coverslips were then washed 3 times with PBS and incubated for 1 h with an Alexa Fluor 488-conjugated goat anti-rabbit IgG antibody (1:1000 dilution, Life Technologies, Thermo Fisher Scientific) in blocking solution, containing 150 ng/mL DAPI (Molecular Probes, Thermo Fisher Scientific) and 100 ng/mL Texas Red-conjugated concanavalin A (Invitrogen, Thermo Fisher Scientific). After three washes with PBS and one with deionized water, coverslips were mounted on glass slides with Mowiol (Sigma-Aldrich, Buchs, Switzerland). Confocal microscopy images were obtained using a Zeiss LSM 710 Meta (Zeiss, Feldbach, Switzerland) at the Cellular Imaging Facility of the University of Lausanne. Additional analyses of the pictures were performed with ImageJ software, version 1.50b (https://imagej.nih.gov). For the calculation of the infection rate, the number of cells and inclusions per field was calculated (200 cells per dilution of the inoculum and per treatment were counted).

### 2.4. Cell Viability Assay

Viability of HEp-2 cells treated with BPDL was assessed using an in vitro resazurin-based toxicology assay kit (Sigma-Aldrich, Switzerland). HEp-2 cells were seeded at 2.5 × 10^4^ cells per well in 96-well plates and grown overnight. The culture medium was then replaced with fresh medium containing increasing concentrations of BPDL and the cells were incubated for 24 h at 37 °C and 5% CO_2_. The viability assay was performed according to the manufacturer’s protocol. Fluorescence was monitored using a FLUOstar Omega microplate reader (BMG Labtech, Offenburg, Germany) at 580 nm, using an excitation wavelength of 540 nm. Untreated cells represented the positive control (100% viability), whereas the negative control was obtained by treating the cells with ice-cold methanol for 5 min. Each condition was tested in triplicate.

### 2.5. RNA Extraction and Sequencing

Per replicate, two 25 cm^2^ flasks containing *W. chondrophila* infected HEp-2 cells were harvested by scraping. Three biological replicates were performed for each condition (untreated RBs, untreated EBs and BPDL-treated). The cell suspension was centrifuged at 1790× *g* for 10 min. The supernatant was then transferred to a new tube and centrifuged at 5000× *g* for 15 min. Both pellets were pooled together in 2.5 mL of TRIzol (Invitrogen), vortexed and stored at −80 °C. RNA extraction was performed according to the manufacturer’s instructions. RNA was resuspended in 60 μL of water and DNA contamination was removed from the samples with the DNA-free^TM^ DNA removal kit (Invitrogen) following the manufacturer’s instructions. To verify the quality, 2 μL RNA were analyzed by Fragment Analyser using the standard sensitivity RNA analysis kit (Agilent Technologies, Santa Clara, CA, USA). Removal of rRNA was performed with the Ribo-Zero rRNA Removal Kit according to the manufacturer’s instructions (Illumina, San Diego, CA, USA). RNA samples were quantified with the Qubit^TM^ RNA Broad-Range Assay Kit (Thermo Fisher Scientific) and sent to Fasteris SA (Geneva, Switzerland), which performed negative selection of Poly-adenylated transcripts (to remove host mRNA), library preparation (Illumina TruSeq Stranded mRNA Library Prep Kit) and sequencing on Illumina NovaSeq 6000. 

### 2.6. Sequence Read Processing and Gene Expression Analysis

Raw reads were trimmed with trimmomatic [53] version 0.36 (parameters: ILLUMINACLIP:NexteraPE-PE.fa:3:25:6, LEADING:28, TRAILING:28 MINLEN:30). Trimmed reads were mapped to the genome of *W. chondrophila* WSU 86-1044 (assembly accession: GCF_000092785.1) with bwa-mem version 0.7.17 (default parameters, [54]). Htseq version 0.11.2 was used to count reads aligned to each *W. chondrophila* gene (parameters: stranded = no t gene). A rarefaction analysis was performed to evaluate if the sequencing depth was sufficient for further analysis: sequencing reads of each sample were downsampled from 500,000 reads up to the total number of reads available by steps of 500,000 reads. Each subsample was analyzed to determine the number of *W. chondrophila* genes detected with progressive increase of the number of reads. The number of detected genes is expected to reach a plateau when the sequencing depth is sufficient to capture all genes expressed under the considered condition.

EdgeR version 3.26.5 [55] was used to perform the differential expression analysis. Library sizes were normalized using the calcNormFactors function with the default “Trimmed Mean of M-values” method [56]. Genes with low read counts were filtered with the edgeR function filterByExpr with default parameters. Dispersion was calculated using the estimateCommonDisp and estimateTagwiseDisp functions. The function exactTest was used to identify genes significantly differentially expressed. Expression changes presenting a false discovery rate corrected *p*-value (FDR) lower than 0.01 where considered significant. Normalized expression values were calculated as Reads Per Kilobase of transcript per Million mapped reads (RPKM). Heatmaps were done with the pheatmap R package (“complete” clustering method) [57] based on log2 transformed RPKM values. KEGG annotations were retrieved using the KEGG API [58]. COG annotations and orthology information were retrieved from the ChlamDB website [59]. GO annotations were retrieved from the Gene Ontology Annotation (GOA) Database (UniprotKB version 2020-02-24) [60]. Gene set enrichment analysis were performed using goatools [61].

## 3. Results

### 3.1. BPDL Treatment Induces the Formation of Aberrant Bodies in W. chondrophila

BPDL is able to chelate both ferric and ferrous species of iron and cause the formation of ABs in *C. trachomatis* [32,33]. Furthermore, BPDL treatment proved more efficient than other compounds (such as deferoxamine mesylate) to induce the transcriptional response to iron starvation, and therefore is more suitable for transcriptomic analysis [33]. As due to the presence of efflux pumps in amoebae the treatment with drugs often requires higher concentrations than in mammalian cells, we chose to grow *W. chondrophila* in human (HEp-2) cells for the RNA-seq experiment. Moreover, we had already observed that BPDL induced the formation of ABs in *W. chondrophila* grown in Vero cells [29]. Since BPDL has never been tested on HEp-2 cells infected with *W. chondrophila*, AB formation was assessed at different concentrations by immunofluorescence microscopy (Figure 1). The treatment with BPDL up to 100 μM did not significantly affect host cells viability (Appendix A). In the case of infected cells, in the presence of 50 μM BPDL bacteria inside the inclusions showed morphologies similar to the untreated control. Significant bacterial enlargement and reduced number of bacteria inside inclusions were observed by increasing the concentration of BPDL (Figure 1). Therefore, only BPDL concentrations above 75 μM were selected for subsequent experiments.

### 3.2. Optimization of the Conditions for the RNA Sequencing Upon BPDL Treatment

Due to *W. chondrophila* obligate intracellular life cycle, the majority of the mRNAs present in the samples originates from the host cells. To obtain a quantity of ABs RNA sufficient for sequencing, two conditions should be fulfilled: (i) high percentage of infected cells and (ii) inclusions mostly filled with ABs. Preliminary experiments were carried out in order to optimize both infection and BPDL treatment protocols. First, the optimal dilution of the *W. chondrophila* inoculum was determined by immunofluorescence. A dilution of 1:100 of the filtered co-culture (as described in the Methods), resulting in 50% of infected cells, was selected (Appendix A, Control). Second, different concentrations of BPDL and durations of treatment were compared (Figure 1 and Appendix A). Based on these analyses, the following protocol was selected: treatment of *W. chondrophila* infected HEp-2 cells with 75 μM BPDL at 8 hpi and cells harvesting for RNA extraction 24 hpi (Figure 2A). The transcriptional response to iron deprivation was monitored comparing BDPL-treated to untreated *W. chondrophila* infected HEp-2 cells, also collected at 24 hpi (Figure 2B). Treatment with BPDL did not affect the infection rate (Appendix A). Immunofluorescence pictures confirmed that the selected protocol for BPDL treatment resulted in the formation of numerous ABs (Figure 2C).

### 3.3. Comparative Transcriptomic Analysis of Persistent W. chondrophila Induced by Iron Starvation

To monitor the transcriptional response to iron starvation in *W. chondrophila*, RNA-seq was performed on BPDL-treated and untreated samples; BPDL samples were harvested 24 hpi, whereas untreated samples were harvested 24 or 72 hpi (RBs and EBs, respectively). Three biological replicates were used for each condition. Between 1.21% and 42.19% of the reads could be mapped to the reference *W. chondrophila* genome (Appendix A). A rarefaction/saturation analysis showed that the number of expressed genes detected by RNA-seq reached a plateau for all samples and replicates in the case of RBs and EBs (24 and 72 hpi, respectively, in Appendix A). In contrast, the three BPDL-treated samples exhibited lower sequencing depth and a higher proportion of host sequences compared to control samples, and their curves did not reach a plateau. The libraries corresponding to samples BPDL-2 and BPDL-3 were therefore subjected to a second sequencing run. The number of reads obtained for sample BPDL-1 during the first sequencing run was so low that this sample was removed from the analysis. Re-sequenced samples exhibited much higher sequencing depth and reached a plateau on rarefaction curves (Appendix A). Samples retained for the analysis had between 5.4 and 9.7 million reads mapping to *W. chondrophila* genome (Appendix A).

The untreated samples harvested 24 and 72 hpi exhibited highly correlated expression profiles (Pearson correlation coefficient >0.99) (Figure 3A). The correlation of the expression profiles of the two BPDL-treated samples was slightly lower (Pearson correlation coefficient of 0.9839) and the samples were relatively distant on the y-axis of the principal component analysis plot as compared to the replicates of RBs and EBs (Figure 3B).

In response to iron deprivation, 32.9% of the *W. chondrophila* genome (657/1′999 open reading frames (ORFs)) was significantly differentially expressed (FDR ≤ 0.01). Among these genes, 325 were down-regulated and 332 up-regulated in comparison to the 24 hpi untreated control (Appendix A; Appendix A). Of these 657 differentially expressed ORFs, 73 (11.11%) are specific to *W. chondrophila*, a proportion similar to that of the complete genome (12.04%). Less than half of the differentially expressed genes (275/657, 41.86%) have at least one homolog in *C. trachomatis*. Many of the differentially expressed genes encode hypothetical proteins of unknown function, as reflected by the fact that 52.5% (345/657) could not be assigned to a COG category of known function (Figure 4). This proportion also corresponds to the proportion of proteins of unknown function encoded in the *W. chondrophila* genome (52.5%). Genes encoding proteins of unknown function were more frequently up-regulated (56.4%) than down-regulated (43.6%) upon BPDL treatment.

Iron deprivation affected numerous biological pathways: genes related to transcription and translation were mainly up-regulated, whereas genes implicated in energy production, carbohydrate transport and metabolism, amino acid transport and metabolism as well as cell wall/envelope biogenesis were generally down-regulated (Figure 4; Appendix A). 

Expression of 56 genes was reduced at least twofold in response to iron starvation **(Appendix A** and Appendix A). Among these genes, *cyoA*, *cyoB* and *cyoC* encode for subunits of the cytochrome ubiquinol oxidase and indicate a down-regulation of oxidative phosphorylation. On the other hand, the expression of 88 genes was increased at least twofold in response to iron starvation (Appendix A and Appendix A). This group includes *mdtA* and *mdtC*, part of the putative multidrug efflux pump *mdtABC*, and *groEL1*, *groES1* and *groES3*, encoding for the GroEL/ES chaperonin machine, which plays an essential role in protein folding [62]. Two putative transcriptional repressors, the heat shock-repressor *hrcA* and *wcw_0478*, were also up-regulated in *W. chondrophila* upon iron starvation. In addition, three up-regulated genes, *rplU*, *rplM* and *rpmE2*, encode for ribosomal proteins.

The volcano plot shown in Figure 5 displays the most down-regulated (Log2FC ≤ −2) or up-regulated genes (Log2FC ≥ 2) exhibiting a minimum of 4-fold decrease or increase, respectively. Additional information concerning these 17 genes is summarized in Table 1. 

Only one gene, *wcw_0483*, annotated as a transposase, showed a major (4-fold) decrease in RNA levels in response to iron deprivation, while 16 genes were up-regulated more than 4-fold. Among them, *wcw_p0002*, *wcw_1348* and *mazF* encode putative toxins belonging to type II toxin-antitoxin systems, suggesting a possible role for such systems in persistence of *Chlamydia*-related bacteria. Interestingly, Wcw_0479, annotated as a putative regulator of cell morphogenesis, is a di-iron-containing protein with homology to *Escherichia coli* YtfE, involved in the repair of iron-sulfur clusters.

## 4. Discussion

As obligate intracellular bacteria, chlamydiae are strictly dependent on iron availability, thus iron deprivation blocks the bacterial developmental cycle. Indeed, we confirmed that iron deprivation induces the formation of ABs in HEp-2 cells infected with *W. chondrophila.* The presence of ABs following iron depletion was previously reported in *C. trachomatis* and *C. pneumoniae* [30,32,33]. To date, the regulation of iron homeostasis in *W. chondrophila* has not been investigated, but the underlying mechanisms might be similar to those previously described in *Chlamydia* species [38].

While numerous stimuli inducing ABs formation in vitro are well described, many aspects of the mechanisms underlying their biogenesis remain unclear. Previous studies investigated the expression of chlamydial genes upon interferon-γ or penicillin treatment and iron depletion by microarray [63,64] or qRT-PCR [65,66,67], and the early response to iron starvation was investigated by RNA-seq (3–6 h of BPDL treatment, before the formation of ABs) in *C. trachomatis* [46]. However, the genome-wide transcriptional activity of ABs in *Chlamydia*-related bacteria was never assessed so far. In order to obtain a complete picture of the transcriptional landscape in *W. chondrophila* ABs, we compared RNA levels of BPDL-treated and untreated infected HEp-2 cells at 24 hpi (treatment protocol in Figure 2A).

We observed a higher variability in gene expression between the two BPDL replicates than between EBs and RBs (Figure 3), which suggests a more heterogeneous transcriptional landscape in *W. chondrophila* ABs compared to untreated EBs and RBs. Further work will be required to assess whether this variability is specific of BPDL-induced ABs or it also occurs upon other stressful conditions. In *W. chondrophila* ABs induced by BPDL treatment, 32.9% of the genome (657/1′999) was differentially expressed as compared to the untreated control (RBs), including 325 (16.3%) down-regulated and 332 (16.6%) up-regulated genes. In comparison, only 7% to 8% of the *C. trachomatis* genome was shown to be differentially expressed upon iron depletion [46]. The comparison between the RNA-seq datasets shows that only 3 homologous genes were differentially expressed in both *C. trachomatis* [46] and *W. chondrophila*: *CTL0332*/*wcw_1181* (encoding SmpB, required for the rescue of stalled ribosomes mediated by trans-translation), *CTL0132*/*wcw_1554* (encoding a protein containing a KH domain, known to bind RNA or ssDNA in proteins associated with transcriptional and translational regulation) and *CTL0575*/*wcw_0582* (*infA*, encoding the essential translation initiation factor IF-1). Nevertheless, these genes were down-regulated by BPDL treatment in *C. trachomatis* and up-regulated in *W. chondrophila*. Similarly, genes related to translation and ribosome translation were also down-regulated upon iron starvation in *C. trachomatis* [46] and up-regulated in *W. chondrophila* (Figure 4), which could represent a major difference between *Chlamydiaceae* and *Chlamydia*-related species. However, as stated above, *C. trachomatis* transcriptional response was investigated after 3 and 6 h of exposure to BDPL, before the formation of ABs. In contrast, the transcriptional response of *W. chondrophila* to iron deprivation was assessed after 16 h of exposure to BPDL, when *W. chondrophila* exhibited the typical shape of persistent chlamydiae (Figure 2C). Therefore, the differences in the transcriptional response to iron depletion between *C. trachomatis* and *W. chondrophila* might reflect mainly the different protocols used for the BPDL treatment. In addition, differences in metabolic capacities, reflected by the large difference in genome size (1.04 Mbp for *C. trachomatis* versus 2.11 Mbp for *W. chondrophila*), could also contribute to the differences observed in the transcriptomic profiles [48].

Our data indicate a significant decrease in expression of genes related to energy production, carbohydrate and amino acid metabolism and cell wall/envelope biogenesis in ABs, compared to actively replicating RBs, which is consistent with a growth arrest induced by stressful conditions. However, most of the differentially expressed genes encode proteins of unknown function. Interestingly, some of these hypothetical proteins are found exclusively in *W. chondrophila*, suggesting that species-specific functions allowing survival in adverse conditions evolved in this obligate intracellular species. Moreover, the genome of *W. chondrophila* exhibits numerous repeated sequences and transposases, in contrast to *Chlamydia* species that almost completely lack these elements [48]. Between 40% and 50% of the genes showing at least a twofold up or down-regulation following iron depletion in *W. chondrophila* are actually in proximity of putative transposases, and three putative transposases are among the most down-regulated genes (*wcw_0483*, *wcw_1956*, *wcw_0001*; Appendix A and Appendix A). This suggests that elements of phage origin or mobile genetic elements could be (partially) responsible for the high proportion of genes of unknown function differentially expressed upon BPDL treatment in *W. chondrophila* (Figure 4).

Among the 88 genes exhibiting at least a 2-fold expression increase upon iron depletion, only *wcw_0479* seems directly related to iron starvation, as it encodes a homolog of YtfE, which in *E. coli* is involved in the biosynthesis of iron-sulfur clusters. Furthermore, *ytfE* was previously shown to be significantly up-regulated under nitrosative stress [68]. Moreover, when the gene is mutated, *E. coli* is more sensitive to iron starvation [69]. Analogously to what observed in *E. coli*, the stress induced by BPDL treatment led to a significant increase of *ytfE* expression in *W. chondrophila*. On the other hand, in both BPDL-treated and untreated samples we observed a high expression of the *ytgABCD* operon (*wcw_1670-wcw_1673*), encoding for the only known ABC transporter implicated in iron acquisition in chlamydiae. The transcription of this operon was not significantly affected by BPDL treatment. This is consistent with previous data reporting a high expression of *C. trachomatis ytgABCD* operon in both untreated and iron-depleted conditions [46].

Interestingly, two genes implicated in oxidative stress, encoding the KatA catalase (*wcw_0655*) and peroxiredoxin 1 (*wcw_0039*), were significantly down-regulated upon iron starvation. Recently, downregulation of genes related to oxidative stress has also been observed in *Burkholderia cenocepacia* upon iron depletion [70], which could represent an iron-sparing measure in iron-limited conditions, as many proteins involved in the oxidative stress response contain iron.

Few genes have been proposed as molecular markers of persistence for *Chlamydiae* species. Among them, the early gene *euo* (*wcw_1683*), identified as a transcriptional repressor regulating *Chlamydia* late genes [71], was up-regulated in persistent *C. trachomatis* induced by BPDL [33]. Increased expression of *euo* upon iron depletion was also reported in *C. pneumoniae* and, to a weaker extent, in *Chlamydia psittaci* [65]. In addition, the up-regulation of *euo* was observed in *C. trachomatis*, *C. pneumoniae* and *C. psittaci* treated with interferon-γ [63,72]. In *W. chondrophila*, *euo* was significantly up-regulated by BPDL treatment in comparison to the untreated control, but showed only a 1.85-fold increase. Interestingly, two other putative transcriptional repressors, *hrcA* and *wcw_0478*, were also up-regulated by BPDL treatment. As shown in Table 1, *wcw_0478* was among the genes exhibiting the highest increase in expression (5.7-fold).

Besides *euo*, the late gene *omcB*, encoding the outer membrane complex protein B, is also considered as a molecular marker of persistence. *omcB* is down-regulated in persistent *Chlamydia* induced either by iron deprivation, interferon-γ or azithromycin [33,63,72,73,74]. In contrast, OmcB protein levels were unaffected in iron-starved *C. pneumoniae* [75]. A down-regulation of *omcB* (*wcw_1162*) in persistent *W. chondrophila* was observed in the current study, although below the 2-fold threshold (1.82-fold decrease). Our data are thus overall consistent with previous studies, which reported an up-regulation of *euo* and a down-regulation of *omcB* in persistent *Chlamydiae*. However, finding common markers of persistence for the different members of the *Chlamydiae* phylum is challenging, since numerous discrepancies are observed between different studies. Indeed, expression levels of putative markers greatly vary according to bacterial species, host cell lines, treatment applied and time-point used for the experiments [76].

In response to iron deprivation, some genes were highly up-regulated or down-regulated (Log2FC ≤ −2 and ≥2). Among the most up-regulated genes, expression of *wcw_p0002*, *mazF* (*wcw_p0022*) and *wcw_1348* was increased 18.7, 5.1 and 4.1 folds, respectively. Interestingly, all three genes encode putative toxins of type II toxin-antitoxin systems. Indeed, both Wcw_p0002 and Wcw_1348 exhibit a HigB-like domain (interpro accession IPR009241), whereas Wcw_p0022 encodes a putative MazF-like toxin, part of the MazEF toxin-antitoxin system [48]. Toxin-antitoxin systems have been described to play a role in bacterial persistence, for example in the case of *E. coli* or *Mycobacterium tuberculosis* [77,78,79]. Indeed, transcriptomic analysis showed that at least 10 toxin-antitoxin systems were significantly up-regulated in *M. tuberculosis* persisters induced by antibiotics [80]. In addition, the deletion of toxin-antitoxin operons was shown to significantly impact the level of *E. coli* persistence [81]. In contrast to *W. chondrophila*, no toxin-antitoxin systems have been identified in *Chlamydiaceae* species, which is likely due to their reduced genomes [82].

## 5. Conclusions

The transcriptional response of obligate intracellular chlamydiae to iron starvation is complex and leads to the development of aberrant bodies, which might represent a persistent form in adverse conditions. Moreover, different stressful conditions seem to induce distinct transcriptional responses. This study represents the first investigation of the transcriptional landscape of aberrant *W. chondrophila* induced by iron starvation and suggests a role of toxin-antitoxin modules in the biogenesis of iron-depleted aberrant bodies. Undoubtedly, a better understanding of the molecular mechanisms leading to the development of persistent forms in *Chlamydiales* will be of extreme importance for the treatment of chronic chlamydial infections.

## Figures and Tables

**Figure 1 microorganisms-08-01848-f001:**
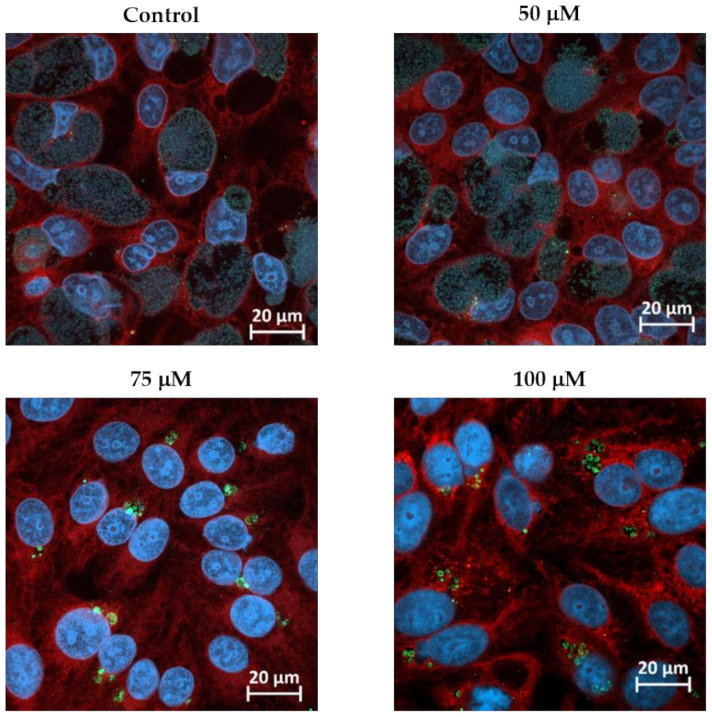
Formation of *W. chondrophila* aberrant bodies in presence of different concentrations of BPDL. HEp-2 cells were infected with *W. chondrophila* and treated with 50, 75 or 100 μM BPDL 2 hpi. Infected cells were fixed 24 hpi, labelled with Concanavalin A (red), DAPI (blue) and anti-*W. chondrophila* antibodies (green) and observed by confocal microscopy. Images of HEp-2 cells infected with *W. chondrophila* and treated with BPDL at different times pi are shown in Appendix A.

**Figure 2 microorganisms-08-01848-f002:**
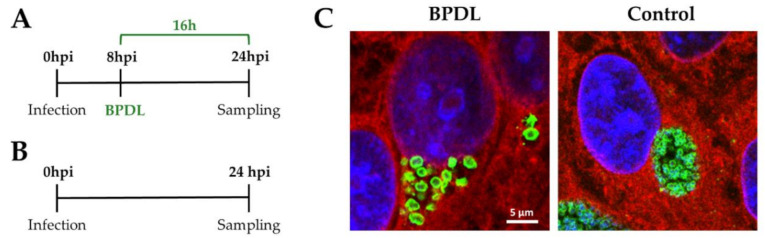
Formation of *W. chondrophila* aberrant bodies in the conditions used for the RNA-seq experiment. (**A**) Protocol for the treated samples (ABs): BPDL was added 8 hpi at a concentration of 75 μM and samples were harvested 24 hpi. (**B**) For the untreated control (RBs), infected HEp-2 cells were harvested 24 hpi. (**C**) In parallel to the RNA-seq experiment, *W. chondrophila* infected HEp-2 cells were grown on glass coverslip in presence (left panel) or absence (right panel) of 75 μM BPDL added 8 hpi. Infected cells were fixed 24 hpi, labelled with Concanavalin A (red), DAPI (blue) and anti-*W. chondrophila* antibodies (green) and observed by confocal microscopy to confirm the formation of ABs. Both pictures display the same scale.

**Figure 3 microorganisms-08-01848-f003:**
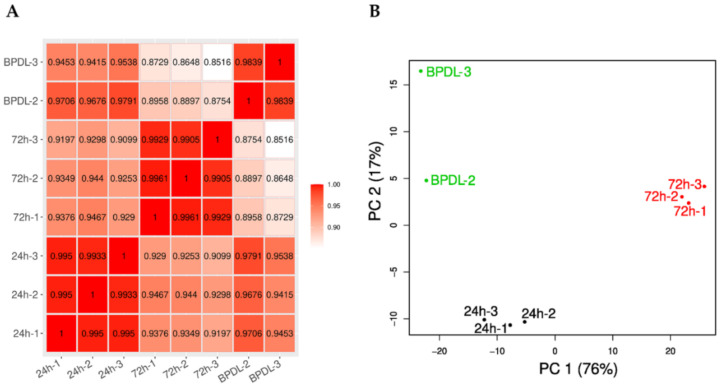
Inter- and intragroup variability of gene expression. (**A**) Pearson correlation of log2 transformed read counts. BPDL-2 and BPDL-3 exhibit lower correlation than the two other conditions (24 and 72 hpi). (**B**) Gene expression principal component analysis (first two dimensions). Replicates for untreated samples (24 and 72 hpi) exhibit lower within-group distances as compared to the two BPDL-treated samples.

**Figure 4 microorganisms-08-01848-f004:**
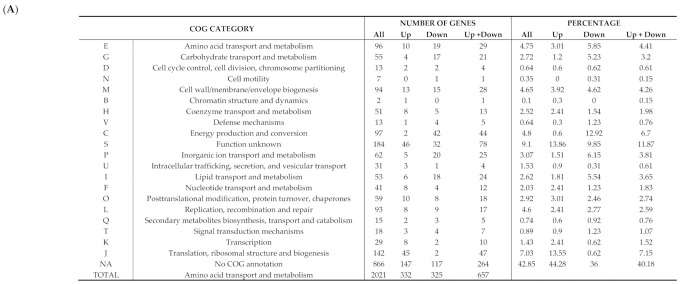
Analysis of differentially expressed genes (BPDL vs. untreated RBs) by functional COG categories. (**A**) The list shows for each COG category: the number of genes (annotated on *W. chondrophila* genome) belonging to the given COG category (All); the number of genes up-regulated by BPDL treatment (Up); the number of genes down-regulated by BPDL treatment (Down); the number of genes up- or down-regulated (Up + down). As the same gene can be assigned to more than one COG category, the total number of genes in the last line exceeds that of the genes encoded by *W. chondrophila* genome. For each number, the fraction (with respect to the total, indicated in the last line) was also calculated. (**B**) The fraction for each COG category shown in A. The graph clearly shows that among the genes up-regulated by BPDL treatment, unknown functions and functions related to translation are over-represented, while energy production, carbohydrate metabolism, amino acid metabolism, inorganic ions transport and metabolism functions are over-represented among genes down-regulated by iron depletion.

**Figure 5 microorganisms-08-01848-f005:**
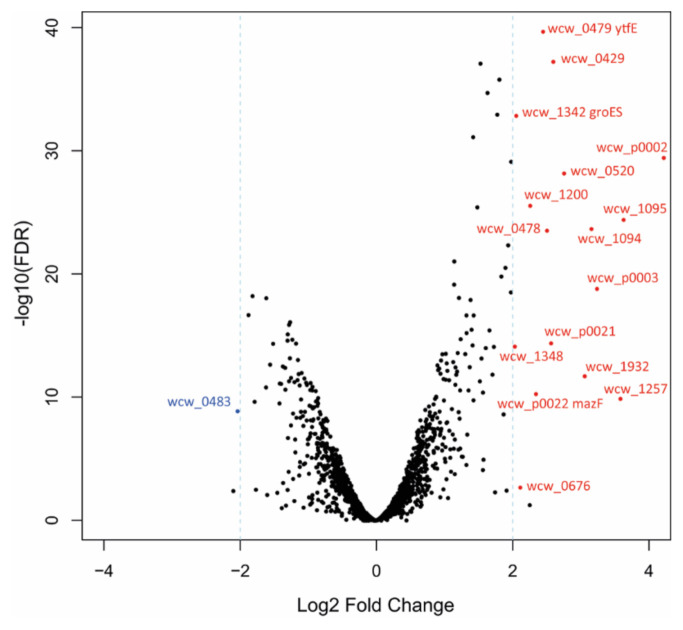
The most down- or up-regulated genes in persistent *W. chondrophila*. Volcano plot displaying the differential gene expression of persistent versus untreated *W. chondrophila*. Each point represents the average value of one gene from two (BPDL-treated) or three (untreated) replicates. Colored points represent the most significantly differentially expressed genes, with an FDR ≤ 0.01 (False Discovery Rate) and a log2FC ≤ −2 (blue) or ≥ 2 (red). If available, the gene name is indicated.

**Table 1 microorganisms-08-01848-t001:** Summary of the genes showing the highest change in expression upon BPDL-treatment.

Gene ^1^	Locus tag	LogFC	FC	FDR	Product	Conservation ^2^
*wcw_0483*	wcw_0483	−2.0	0.2	2.03 × 10^−8^	Transposase	B
*wcw_1348*	wcw_1348	2.0	4.1	3.15 × 10^−13^	Putative addiction module killer protein	B
*groES1*	wcw_1342	2.1	4.2	3.69 × 10^−31^	Co-chaperonin GroES	A
*wcw_0676*	wcw_0676	2.1	4.3	0.0064	tRNA-Ser	A
*wcw_1200*	wcw_1200	2.3	4.8	4.31 × 10^−24^	Hypothetical protein	C
*mazF*	wcw_p0022	2.4	5.1	1.01 × 10^−9^	Putative MazF-like toxin	B
*wcw_0479*	wcw_0479	2.5	5.5	3.82 × 10^−37^	Putative regulator of cell morphogenesis	B
*wcw_0478*	wcw_0478	2.5	5.7	3.39 × 10^−22^	Putative HTH-type transcriptional repressor	B
*wcw_p0021*	wcw_p0021	2.6	6.0	1.88 × 10^−13^	Hypothetical protein	B
*wcw_0429*	wcw_0429	2.6	6.1	4.94 × 10^−35^	Hypothetical protein	C
*wcw_0520*	wcw_0520	2.8	6.8	1.11 × 10^−26^	Hypothetical protein	B
*wcw_1932*	wcw_1932	3.1	8.4	4.90 × 10^−11^	tRNA-Gly	A
*wcw_1094*	wcw_1094	3.2	9.0	2.64 × 10^−22^	Hypothetical protein	B
*wcw_p0003*	wcw_p0003	3.2	9.5	1.30 × 10^−17^	Hypothetical protein	B
*wcw_1257*	wcw_1257	3.6	12.0	2.35 × 10^−9^	tRNA-Ala	A
*wcw_1095*	wcw_1095	3.6	12.4	5.10 × 10^−23^	Hypothetical protein	B
*wcw_p0002*	wcw_p0002	4.2	18.7	7.40 × 10^−28^	Putative component of the toxin-antitoxin plasmid stabilization module	B

^1^ The table includes only the genes with a log2FC ≤ −2 or ≥2 and an FDR ≤ 0.01 (in red or blue in Figure 5). ^2^ Refers to available genomes. A: Conserved in all chlamydiae species. B: Conserved in several chlamydia-related species. C: Specific to *W. chondrophila*.

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
