# Peer review of "Transcriptional Landscape of Waddlia chondrophila Aberrant Bodies Induced by Iron Starvation"

_microorganisms, 2020, doi:10.3390/microorganisms8121848_

Round 1

Reviewer 1 Report

The revised manuscript did not address the concern regarding the normalization of the RNA signals. This could have a major impact on all the chlamydial gene expression results. 

Author Response

As mentioned in the Method section, the RNA-seq data was normalized using the “Trimmed Mean of M-values” normalization method (TMM) described in Robinson and Oshlack (2010) using the “calcNormFactors(X)” implemented in the edgeR package.

Other normalization methods (described in Conesa et al., Genome Biology 2016), such as the “reads per kilobase per million reads” (RPKM), are mainly used for within-sample normalization, as they remove feature-length and library-size effects. However, correcting for gene length is not necessary when comparing the expression of the same gene in different samples. In this case, as samples can have heterogeneous transcript distributions (so highly and differentially expressed genes can skew the count distribution), normalization techniques such as TMM are applied. This is also the default normalization method in edgeR (Chen et al., edgeR: differential analysis of sequence read count data – User’s Guide).

For these reasons, we applied the TMM method for normalization. The paper cited by the reviewer (Ouellette et al, Molecular Microbiology 2006) compares two different normalization methods for microarray data, 16S rRNA or genome copies number, concluding that the second is more appropriate as the ratio between RNA and DNA decreases along the chlamydial developmental cycle. This conclusion suggests that using the 16S rRNA for normalizing transcriptional data (such as microarrays or RT-qPCR) might introduce a bias, but in our opinion this is not applicable to our RNA-seq data, as we did not use a specific feature (such as 16 rRNA) for normalization. We understand that the reviewer might have criticisms about the available methods on RNA-seq data normalization, but the purpose of our work was not the evaluation of the existing methods for the normalization of RNA-seq data, a task for which we do not claim to have the necessary expertise. We applied the most widely used protocol for the comparison of transcriptional data in different conditions, based on the literature.

Reviewer 2 Report

The authors have done a nice job addressing previous comments.

The finding that mobile / phage-encoded genetic elements may be driving a large portion of the expression changes for 'genes of unknown function' adds significance to the results.

The description of the methodology and controls used has also been improved.

Author Response

We thank the reviewer for the comments that helped improving the quality of the manuscript.

Round 2

Reviewer 1 Report

The fundamental problem of normalization of the previous version of the manuscript was not addressed in the revision. The authors need to reanalyze the data using normalized Waddlia gene expressions. The normalization in my opinion can be done using the 16S Waddlia rRNA levels. Or if it was removed during the RNA processing, than another previously described Waddlia housekeeping gene should be used. 

This manuscript is a resubmission of an earlier submission. The following is a list of the peer review reports and author responses from that submission.

Round 1

Reviewer 1 Report

Silvia Ardissone et al. described the global transcriptional response of Waddlia chondrophila to a stress factor – iron limitation. While various stress factors have been described to lead to persistence of chlamydial species, iron deprivation mediated persistence has not been described at the transcriptome level for Waddlia chondrophila. While the quality of the study is acceptable, there are major questions that should be answered before accepting it.

104-105    Specify the titer of the bacterium.

110-111   Specify the MOI.

226-227 What was the method of normalization of the RNA signals? It is a crucial question, that determines the whole study. Was it normalized to a bacterial housekeeping RNA? Was it normalized to the amount of bacterial DNA (measured by e.g. qPCR)? The bacterial RNA signal is coming from many small bacteria or a few big ABs?

Additional problem arises at rows 226-227 where the authors stated that the BDPL-2 and BDPL-3 samples went through a second sequencing run, while the control samples went through only one sequencing run. It should be explained how can one compare the treated and the control samples?   These are fundamental questions that should be answered, otherwise the paper is not acceptable – we don’t know what gene is upregulated and downregulated. It is useful to read the paper by Bob Belland about this topic (https://pubmed.ncbi.nlm.nih.gov/17059564/).

Figure 3A – borders between the cells would be very useful.

Figure 4+Figure 5 and Figure 6. are redundant figures – may be Figure 5 and 6 can go to supplementary material, and the functional analysis figures (eg. Suppl Figure 3.) should be put in the manuscript. I generally feel that the biological meaning of the gene expression changes should be more discussed throughout the manuscript.  I also have to note that among the supplementary material I could find only the regular figures of the paper – no supplementary figures and tables.

315-317  One cannot make such a conclusion based on two samples.

Reviewer 2 Report

The authors utilize an RNA-sequencing approach to compare the transcriptional profile of ABs induced by iron starvation to untreated bacteria in the Chlamydia-related species Waddlia chondrophila. They conclude that their results indicate a significant reduction in the expression of genes related to energy production, carbohydrate and amino acid metabolism and cell wall/envelope biogenesis, compared to untreated, actively replicating bacteria. Conversely, three putative toxin-antitoxin modules were among the most up-regulated genes upon iron starvation, suggesting that their activation might be involved in growth arrest in adverse conditions, an uncommon feature in obligate intracellular bacteria. This is the first report of a complete transcriptomic profile of a Chlamydia-related species in stressful conditions and has implications for answering questions about the evolutionary development of stress response and persistence in Chlamydia and Chlamydia-related species.

This work is interesting, and its value is greatest when compared to similar studies conducted on other members of the Chlamydiales, specifically Chlamydia trachomatis. However, there is a concern about the methodology used as well as the interpretation of some of the results. The authors report large differences within their treatment group, when compared to their two control conditions. They also tend to highlight the similarities with previously published works on members of the Chlamydiales (C. trachomatis and C. pneumoniae) while going into far less detail on the differences they have found. The most striking being the apparent upregulation of genes associated with protein synthesis, which is exactly the opposite of what occurs in C. trachomatis during iron-induced persistence.

Major Comments:

Supplementary Tables / Figures were not provided. The supplemental zip file contained only the primary figures for the manuscript. This needs to be addressed before subsequent review.

The authors should state why they chose to carry out their RNA-seq experiments in Hep-2 cells, rather than A. castellanii. Are ABs observed when Waddlia is grown in A. castellanii? How dramatically does the transcriptional profile of Waddlia differ when cultured in A. castellanii and Hep-2 cells?

 Figure 1: Because the conditions have not been previously reported for Hep-2 cells and W. chondrophila (as the authors state), it would seem prudent to conduct cell cytotoxicity assays at the various conditions used in Figure 1. This would eliminate the possibility that the observed physiological changes (particularly at higher concentrations of BPDL) in W. chondrophila were not due to indirect effects.

 Lines 202-205: Because the conditions used by the authors have not been previously reported and will directly affect the outcome of their experiments, the authors need to better explain why they chose to use the conditions that they did. Data showing the results of different treatment conditions would be preferable, but if ‘data not shown’ is used, the authors need to at least explain the actual results they obtained when alternative conditions were used and then defend their preferred methodology. As an example, the authors decided to add their iron chelator at 8hpi. When cultures were treated prior to 8 hpi, what was the result? Loss of infectivity? Cell death / apoptosis? No effect on either bacterium or host cell viability? This information is necessary for the readers to validate the experimental design and the interpretation of the results.

Line 222-226: This is highly informative. How well does this track with what has been published in C. trachomatis? Did the authors consider trying to overcome this obstacle by infecting their cells at a higher MOI thus enhancing the overall abundance of DNA and subsequent transcription? This is why it would be beneficial to know how and why the authors decided on their preferred infection and treatment methods.

While DNA replication continues during most aberrance-inducing conditions, others studies in C. trachomatis have reported that it does so at a reduced rate, particularly later in the developmental cycle. How do the authors know that the reduced transcription they see is not simply the result of less template DNA to transcribe? It would seem that a control would be to measure chromosomal copy number at the time point in the developmental cycle that the authors performed their RNA-seq analysis for the untreated and iron-chelated conditions, to ensure that equal numbers of chromosomes were present at the time point they conducted their study. This would allow them to determine the degree to which reduced chromosomal copy number was responsible for the corresponding effects on transcription.

Line 233-236: How does this affect the interpretation of the results? Given the close association of the other two conditions (untreated 24 and 72 hours), how much confidence do the authors have that the data presented in their treatment group is representative? It may be prudent for the authors to confirm independently that the genes they associated as being up or down – regulated were found to be as such for both BPDL-2 and BPDL-3 groups independently. Otherwise, general conclusions drawn from this dataset run the risk of being the result of over-interpretation.

Minor Comments:
Are the toxin-antitoxin genes associated with proximal phage elements? This could largely explain their activation under the stress-induced condition of iron deprivation.

Line 84-85: This statement requires a citation or two.

Line 244-246: It would be helpful to know something about the developmental cycle of Waddlia to digest this result. For example, of the genes that appear upregulated, do they tend to be those normally associated with the beginning, middle, or end of the developmental cycle? Because gene expression is largely temporally regulated in Chlamydia species, this could be due to a block in development, enabling ‘early’ genes to be expressed longer while ‘mid’ genes could be delayed.

Line 254-255: genes related to transcription and translation were mainly up-regulated”

This is noteworthy, as in C. trachomatis, genes related to translation are downregulated as a result of iron sequestration. When coupled with “three up-regulated genes, rplU, rplM and rpmE2, encode for ribosomal proteins” (lines 265-266), this would appear to be a major difference between iron stress in Chlamydia and Chlamydia-like organisms. The authors should highlight this point, as it appears to be a major difference between Chlamydia and Chlamydia-like organisms.

Figure 4 and 5: Multiple ‘transposases’ are listed in Figure 4,5. It might be prudent to screen the gene categories for genetic evidence of other indicators of potential phage origin and/or mobile genetic elements. This could explain why so many regions of unknown function appeared to be upregulated, as genes of phage origin often respond to stress-inducing conditions.